# Anti-Proliferative and Cytoprotective Activity of Aryl Carbamate and Aryl Urea Derivatives with Alkyl Groups and Chlorine as Substituents

**DOI:** 10.3390/molecules27113616

**Published:** 2022-06-04

**Authors:** Maxim Oshchepkov, Leonid Kovalenko, Antonida Kalistratova, Maria Ivanova, Galina Sherstyanykh, Polina Dudina, Alexey Antonov, Anastasia Cherkasova, Mikhail Akimov

**Affiliations:** 1Department of Chemistry and Technology of Biomedical Drugs, Mendeleev University of Chemical Technology of Russia, Miusskaya sq. 9, 125047 Moscow, Russia; lkovalenko@muctr.ru (L.K.); kalistratova.a.v@muctr.ru (A.K.); ivanova.ms.rcu@yandex.ru (M.I.); 2Shemyakin-Ovchinnikov Institute of Bioorganic Chemistry, Russian Academy of Sciences, Ul. Miklukho-Maklaya, 16/10, 117997 Moscow, Russia; galya24may@gmail.com (G.S.); polinadudkinz@gmail.com (P.D.); 3Faculty of Mechanics and Mathematics, Lomonosov Moscow State University, GSP-1, Leninskie Gory, 119991 Moscow, Russia; alexey.p.antonov@gmail.com; 4Faculty of Biotechnology, Lomonosov Moscow State University, GSP-1, Leninskie Gory, 119991 Moscow, Russia; a-cherkasova2000@mail.ru

**Keywords:** carbamates, oxamates, synthetic cytokinins, substituted ureas, anti-stress effect, cytotoxicity, oxidative stress

## Abstract

Natural cytokinines are a promising group of cytoprotective and anti-tumor agents. In this research, we synthesized a set of aryl carbamate, pyridyl urea, and aryl urea cytokinine analogs with alkyl and chlorine substitutions and tested their antiproliferative activity in MDA-MB-231, A-375, and U-87 MG cell lines, and cytoprotective properties in H_2_O_2_ and CoCl_2_ models. Aryl carbamates with the oxamate moiety were selectively anti-proliferative for the cancer cell lines tested, while the aryl ureas were inactive. In the cytoprotection studies, the same aryl carbamates were able to counteract the CoCl_2_ cytotoxicity by 3–8%. The possible molecular targets of the aryl carbamates during the anti-proliferative action were the adenosine A2 receptor and CDK2. The obtained results are promising for the development of novel anti-cancer therapeutics.

## 1. Introduction

The plant hormones cytokinins are predominantly adenine-derived regulatory molecules that take part in almost all stages of plant growth and development. Studies of the biological activity of cytokinins in animal cells, implemented mainly on 6-substituted purines, in particular on kinetin [1,2], have shown the presence of antiviral, antiparasitic, antitumor, antioxidant and other therapeutic properties [3,4,5,6]. At the same time, many chemical compounds with cytokinin activity other than substituted purines are known, but their activity has not been investigated on animal and human cells.

One of the promising classes of cytokinin-like compounds are aryl and heteroaryl urea derivatives. Some derivatives, such as 1-phenyl-3-(4-pyridyl) urea (4PU), exhibit surprisingly high cytokinin activity in tobacco callus culture. Some synthetic derivatives were even more active than natural endogenous cytokinins [7,8], but studies of other types of biological activity of these compounds were not carried out. Another class of synthetic cytokinin analogs contain urea and carbamate moieties with an ethylene linker. Among them, the oxalylaryl carbamates (Figure 1, Structure **II**) have anti-stress growth-regulatory activity for crops [9,10], and ethylene diurea (EDU, Figure 1, **III**) has ozone protective properties [11,12,13,14].

Cytokinin-like phenylureas bind to the same site as cytokinin receptors [15], are stable, resistant to the action of oxidases, and contribute to an increase in the activity of peroxidase and superoxide dismutase.

Chemical modification of cytokinin analogs may result in both an increase in their pro-proliferative effects and in an inversion of their activity. Thus, it was shown that the introduction of substituents into the aromatic ring increases the activity, and electron-withdrawing substituents lead to a greater effect than electron-donating substituents [16]. Chlorine derivatives of nonpurine analogues of cytokinins usually have a higher proliferative activity [17].

In addition to the cytoprotective effect, the analogs of EDU were shown to exhibit anticancer activity via ROS-dependent apoptosis induction with EC_50_ of about 10–20 µM [18], but the available data on this topic are quite limited. Earlier we synthesized a series of aryl-substituted ureas and carbamates containing aromatic chlorine and a modified imidazolidinone moiety. These compounds were found to be cytotoxic to the breast cancer cell line MDA-MB-231, glioblastoma U-87 MG and neuroblastoma SH-SY5Y, but not to the melanoma A-375 cell line. The introduction of chlorine into the aromatic ring of cytokinin analogues significantly reduced the cytotoxicity, but at the same time provided the capability to protect cells from oxidative stress induced by H_2_O_2_ [19]. The observed cytotoxicity was quite low (EC_50_ of about 100 µM) but, given the scarcity of the data, there was a high probability that there could be more active compounds among other similar cytokinin analogs.

In the current research, we tried to find more active analogs of cytokinins with both anti-cancer and cytoprotective activities. We synthesized a novel set of modifications of 4PU (**I**), EDU (**III**), and oxalylaryl carbamates (**II**) with alkyl and chlorine substitutions and evaluated of their anti-proliferative and cytoprotective activity. Aryl carbamates with the oxamate moiety were anti-proliferative for the cancer cell lines tested, while the aryl ureas were inactive. In the cytoprotection studies, all the derivatives displayed little or no activity. The possible molecular targets of aryl carbamates during the anti-proliferative action were the adenosine A2 receptor and CDK2.

## 2. Results

### 2.1. Compound Synthesis

The preparation of 1-phenyl-3-(4-pyridyl) urea derivatives **I** (Table 1) was carried out accordingly to known methods [20,21]. The last stage consisted of the interaction of phenyl isocyanate with 4-aminopyridine. To convert 4-aminopyridine and 4-amino-2-chloropyridine salts into the free form and accelerate the process, basic catalysis was used by adding a few drops of triethylamine to the reaction mixture.

Aryl ureas and aryl carbamates (Table 1) were produced by the reaction of corresponding aryl isocyanates with imidazolidinone-substituted alcohol or amine in the presence of triethylamine in anhydrous toluene (for aryl ureas) or acetonitrile (for aryl carbamates) as described in the literature [19]. Oxalylaryl carbamates were obtained in the same way as described in refs. [9,10].

### 2.2. Anti-Proliferative Activity Evaluation

We first tested the synthesized compounds for their ability to induce cell death or decrease proliferation in a set of cancer cell lines. We used human cell lines for three major cancer types (glioblastoma U-87 MG, melanoma A-375, metastatic breast cancer MDA-MB-231), and a neuroblastoma SH-SY5Y, which was later intended to be a model in a cytoprotection setting. The cells were incubated with the test compounds overnight, and their proliferation was evaluated using the MTT assay. The compounds were assayed in the concentration range of 1–100 µM to account for the lowest potential load for the patient’s organism.

All of the compounds from the analogs of EDU series **III***a*–*h* displayed no cytotoxicity for all cell lines tested (Figure 2). On the other hand, all aryl carbamates except **II***c* were moderately anti-proliferative for all cell lines, decreasing the cell viability by about 40% at 100 µM (Figure 3). Pyridyl urea derivatives demonstrated low anti-proliferative activity; the most active of them was **I***c* (Figure 4).

### 2.3. Selectivity of the Active Arylcarbamates

To investigate substance selectivity, we used normal immortalized human fibroblast cell line Bj-5ta in the same experimental setting as in the cytotoxicity studies. The compounds displayed slight anti-proliferative activity with about 10% cell death at 100 µM of the substance (Figure 5). The selectivity indices were not calculated because of the very low cytotoxicity of the compounds for the Bj-5ta cell line in the designated concentration range. However, at 100 µM, **II***a*, *b* and *d* compounds induced a 32–42% proliferation decrease in the cancer cell lines and only 4–17% in the Bj-5ta cell line. Based on these data, compound selectivity calculated as the anti-proliferative activity ratio at the 100 µM concentration was 2 to 8 (Table 2).

We chose the substance **II***d* as a model to further evaluate the selectivity and cell death type based on the observed anti-proliferative activity.

### 2.4. Cell Death Type and Mechanism of the Active Arylcarbamates

An investigation of cell death type and mechanism was performed for the **II***d* compound on the MDA-MB-231 cell line. Several sets of experiments were performed: (1) cell staining with DNA binding and phosphatidylserine binding dyes with further microscopy to detect necrosis and apoptosis, accordingly; (2) measurement of caspase 3, 8, and 9 activity; (3) measurement of the ability of blockers of necroptosis (necrostatin-1 and necrosulfonate), apoptosis (Z-VAD-FMK), autophagy (hydroxychloroquine), and of a ROS scavenger (N-acetyl cysteine) to prevent **II***d* cytotoxicity.

**II***d* treatment led to a slight increase in caspase 3 activity and, to some extent, induced phosphatidylserine externalization (Figure 6). Neither of the inhibitors used was able to prevent the cytotoxicity of the compounds, except for the N-acetylcysteine (Figure 7). In accordance with that, the compound induced an increase in the intracellular ROS concentration (Figure 7). However, the inhibition of the ROS-sensitive kinase ASK1 did not reduce the compound’s cytotoxicity. These data indicate that the compound induces both apoptosis and necrosis, and possibly slows down cell proliferation.

### 2.5. Cytoprotection

Based on the data on the cytoprotective activity of the structurally similar compounds [5,19,22,23], we tested the synthesized compounds in two antioxidant models (protection against the H_2_O_2_ and CoCl_2_ cytotoxicity) in a 24 h incubation. In addition, we evaluated the analogs of EDU **III** for their ability to stimulate cell proliferation after a 72 h treatment. We did not test aryl carbamates **II** and pyridyl ureas **I** for this activity, as these compounds were significantly cytotoxic (Figure 3).

In the cytoprotection experiments, the substances were mainly inactive (Figure 8). However, **II***c* and **II***d* at concentrations from 1 to 10 µM were able to increase the cell survival in the CoCl_2_ cytotoxicity test by 3 to 8%.

Among the aryl ureas, **III***f* exhibited a statistically significant pro-proliferative effect at concentrations of 1–10 µM, and **III***b*, *c*, and *d* demonstrated anti-proliferative action at the concentration of 100 µM (Figure 9).

### 2.6. Molecular Docking

Since the compounds demonstrated a substantial anti-proliferative activity with a pro-apoptotic compound, we decided to perform a series of experiments to gain some insights into their molecular targets. We hypothesized that the synthesized compounds and their molecular prototypes cytokinins could share at least some of the receptors.

Cytokinins, the molecular prototypes of the synthesized compounds, have several core molecular targets in mammalian cells: adenosine A2 receptor (A2AR), adenine phosphoribosyltransferase (APRT), and cyclin dependent kinase 2 (CDK2). To probe these proteins as the potential targets for the synthesized compounds, we performed a set of molecular docking experiments using the AutoDock Vina tool. For each protein, several conformation variants were analyzed (A2AR, 5mzj [24], 2ydo [25], and 5mzp [24]; APRT, 6hgs [26], 6hgr [26], and 6hgp [26]; CDK2, 5fp5 [27] and 2jgz [28]).

For A2AR, aryl carbamates **II** typically displayed affinity between the inhibitor caffeine and activator adenosine, while EDU analogs **III** had a lower affinity than both caffeine and adenosine (Figure 10, Table 3, Appendix A).

For APRT, the affinity of all compounds was much lower than for the substrate GMP and inhibitor IMP (Figure 11, Table 4 and Appendix A).

For CDK2, aryl carbamates **II** displayed affinity close to that of the inhibitor SCP2, and aryl ureas had a much lower affinity (Figure 12, Table 5, Appendix A).

## 3. Discussion

In this paper we report the synthesis and evaluation of some novel pyridyl urea, aryl urea and carbamate derivatives with alkyl and chlorine substitutions for biological activity. The compounds were designed to fill the gap in the known synthetic analogs of the substituted cytokinin-like derivatives. This research continues our earlier study [19], extending it with novel compounds and data on the activity mechanisms. The task looked promising as such compounds are known for exerting cytoprotective and antitumor activity.

To synthesize the designated derivatives, we used known literature methods with the yield in the range of 15 to 55%, which is typical for such compound types.

The synthesized compounds were evaluated for their ability to induce cell death in a set of human cancer cell lines (glioblastoma U-87 MG, melanoma A-375, and metastatic breast cancer MDA-MB-231) chosen based on the clinical significance of the corresponding tumors and on the neuroblastoma SH-SY5Y cell line, which was later intended to be used in the cytoprotection tests. EDU analogs **III** derivatives were not toxic up to the concentration of 100 µM (Figure 2) after 24 h of incubation, but **III***b*, *c*, *d* and *g* displayed some anti-proliferative activity after 72 h (Figure 9). However, **III***f* in the latter experiment setting stimulated cell proliferation in SH-SY5Y. Such pro-proliferative activity is quite typical for the cytokinin analogs [5,19,22,23].

Aryl carbamate compounds **II** were anti-proliferative for all cell lines, and three of them demonstrated substantial selectivity compared to the immortalized fibroblast cell line (Table 2). The activity, however, was relatively low for a cytotoxic compound but substantial for an anti-proliferative one, with a 20–40% cell proliferation decrease at a concentration of 100 µM. Pyridyl urea **I***c* was also anti-proliferative, with some preference toward the melanoma and breast cancer cell line (Figure 4). The observed activity was in line with the already described in the literature [29].

Based on the discovered selectivity of the aryl carbamates **II**, we used a set of methods to describe the type of cell death induced by them, with **II***d* as the model compound. We used blockers of necroptosis, apoptosis, and autophagy, stained the cells with the apoptosis-sensitive dye, and evaluated the activation of caspases 3, 8, and 9. **II***d* induced only a slight increase of caspase 3 activity and apoptotic cell staining, and the only blocker able to decrease its cytotoxicity was the ROS scavenger N-acetyl cysteine. The latter’s activity agreed with the **II***d* induced accumulation of ROS in the cells (Figure 7). These results point to the primarily anti-proliferative mechanism of the action of the compounds.

To obtain more insights into the molecular mechanism of action of the aryl carbamates **II** and EDU analogs **III**, we performed molecular docking studies with the most known cytokinine analogs targets: adenosine A2 receptor, ARPT, and CDK2 [30,31]. We observed affinities close to those of the known inhibitors toward the A2AR and CDK2 for compounds **II,** and much lower affinities for the compounds **III** (Table 3 and Table 5). These results agree with the literature data on the anti-proliferative activity of the inhibitors of these proteins [32,33], but a more detailed study is required to prove this interaction.

Based on the literature data on the ability of the cytokinin derivatives to protect cells against various stress, we tested the synthesized compounds for their ability to protect cells against the cytotoxicity of H_2_O_2_ and CoCl_2_, and for their ability to stimulate cell proliferation directly. In these experiments, the substances mainly were inactive (Figure 8). However, **II***c* and *d* at concentrations from 1 to 10 µM were able to increase the cell survival in the CoCl_2_ cytotoxicity test by 3 to 8%. Among the aryl ureas, **III***f* exhibited a statistically significant pro-proliferative effect at concentrations of 1–10 µM.

The obtained data on the aryl urea, aryl carbamate, and pyridyl urea derivatives demonstrated their ability to inhibit cancer cell proliferation. The probable targets of this activity are adenosine A2 receptor and CDK2, but a more detailed study is required to obtain the molecular details of this interaction.

## 4. Materials and Methods

### 4.1. Materials

L-glutamine, fetal bovine serum, penicillin, streptomycin, amphotericin B, Hanks’ salts, Earle’s salts, trypsin, DMEM, MEM, and (4,5-dimethylthiazol-2-yl)-2,5-diphenyltetrazolium bromide (MTT) were from PanEco, Moscow, Russia. 2′,7′-Dichlorodihydrofluorescein diacetate (DCFH-DA), isopropanol, HCl, CHAPS, protease inhibitor cocktail, EDTA, dithiothreitol, HEPES, DMSO, SCP0139, toluene, acetonitrile, carbon tetrachloride, diethylenetriamine, urea, triethylamine, 4-chlorophenyl isocyanate, 3,4-dichlorophenyl isocyanate, and D-glucose were from Sigma-Aldrich, St. Louis, MO, USA. Hydroxychloroquine, Z-VAD-FMK, necrostatin-1, necrosulfonate, NQDI-1, Ac-DEVD-AFC, and Ac-LEHD-AFC were from Tocris Bioscience, Bristol, UK. The apoptosis assay kit was from Abcam, Cambridge, MA, USA.

Cell lines were purchased from ATCC, Manassas, VA, USA.

### 4.2. Synthesized Compounds’ Characterization

Structures of all synthesized compounds were confirmed by ^1^H and ^13^C NMR spectroscopy (Appendix A), mass spectrometry and elemental analysis data. The purity of the compounds was confirmed by HPLC-MS and was in the range of 95–99%. ^1^H and ^13^C NMR-spectra were recorded with a «Bruker DRX-400» spectrometer operating at 400.13 MHz frequency, using DMSO-*d6* as solvent and TMS as an internal standard. Chemical shifts were measured with 0.01 ppm accuracy, coupling constants are reported in Hertz. HPLC-MS was recorded on an inductively coupled plasma mass spectrometer XSeries II ICP-MS (Thermo Scientific Inc., Waltham, MA, USA). Melting point was determined using the melting point (temperature) apparatus Stuart SMP20 (Cole-Palmer, Stone, Staffordshire, UK).

For a qualitative analysis of reaction mixtures compositions, aluminum TLC plates with silica gel (0.015–0.040 mm) with a fluorescent indicator F254 (20 × 20 cm^2^) (Merck Millipore, Darmstadt, Germany) were used. For preparative chromatographic separation of the substances mixtures, «Kieselgel 60» silica gel (0.015–0.040 mm, Merck Millipore, Darmstadt, Germany) was used.

### 4.3. Chemical Synthesis

The preparation of 1-phenyl-3-(4-pyridyl) urea derivatives (**I**) was carried out according to known methods described in the literature [20,21,34]. An amount of 1 eq. of 4-aminopyridine in dry acetone (15 mL per 15 g of substance) was mixed with the solution of relevant phenyl isocyanate in dry acetone (10 mL per 0.5 g of substance). The reaction mixture was held at room temperature for 48 h, then it was concentrated, and the product was purified by column chromatography (silica gel) using Acetone/Chloroform (1:1) followed by crystallization from ethyl acetate.

*N*-(4-pyridyl)-*N’*-phenylurea (**I***a*). 43% yield, mp = 165–167 °C (mp = 162–163 °C [34]). ^1^H NMR (DMSO-*d_6_*, δ, ppm, J, Hz): 6.99 (dt, 1H, -C3H-, J = 7.3, 1.0); 7.23–7.31 (m, 2H, -C2H-C3H-C4H-); 7.40 (dd, 2H, -C12H-C-C13H-, J = 4.8, 1.5); 7.44 (dd, 2H, -C1H-C-C5H- J = 8.5, 1.0); 8.33 (d, 2H, -C14H-N-C16H-, J = 6.1); 8.73 (s, 1H, -NH-Ph); 8.97 (s, 1H, -NH-Py). HPLC-MS: [M + 1]+ found 214.18; calculated value is 214.24.

*N*-(2-chloro-4-pyridyl)-*N’*-2-tolylurea (**I***b*). 54% yield, mp = 189–190 °C (mp = 184–185 °C [34]). ^1^H NMR (DMSO-*d_6_*, δ, ppm, J, Hz): 2.23 (s, 3H, CH3-); 7.00 (dt, 1H, -C3H-, J = 7.4, 1.0); 7.13–7.19 (m, 2H, -C2H-C3H-C4H-); 7.27 (dd, 1H, -C12H-, J = 5.7, 1.9); 7.64 (d, 1H, -C13H-, J = 1.8); 7.68 (d, 1H, -C5H- J = 7.6); 8.12 (s, 1H, -N7H-); 8.14 (d, 1H, -C16H-, J = 5.6); 9.52 (s, 1H, -N9H-). HPLC-MS: [M + 1]+ found 262.26; calculated value is 262.07.

*N*-(2-chloro-4-pyridyl)-*N’*-3-tolylurea (**I***c*). 23% yield, mp = 92 °C (mp = 93–95 °C [34]). ^1^H NMR (DMSO-*d_6_*, δ, ppm, J, Hz): 2.27 (s, 3H, CH3-); 6.82 (d, 1H, -C3H-, J = 7.3); 7.15 (t, 1H, -C4H-, J = 7.7); 7.21 (d, 1H, -C5H-, J = 8.2); 7.28 (m, 1H, -C1H-); 7.29 (dd, 1H, -C12H-, J = 5.7, 1.9); 7.62 (d, 1H, -C13H-, J = 1.8); 8.14 (d, 1H, -C16H-, J = 5.6); 8.77 (s, 1H, -N7H-); 9.22 (s, 1H, -N9H-). HPLC-MS: [M + 1]+ found 262.26; calculated value is 262.07.

*N*-(2-chloro-4-pyridyl)-*N’*-3-chlorophenylurea (**I***d*). 14% yield, mp = 199–200 °C (mp = 198–199 °C [34]). ^1^H NMR (DMSO-*d_6_*, δ, ppm, J, Hz): 7.04 (dt, 1H, -C3H-, J = 6.8, 2.1); 7.27–7.33 (m, 3H, -C4H-, -C5H-, -C12H-); 7.60 (d, 1H, -C13H-, J = 1.8); 7.64 (t, 1H, -C1H-, J = 1.8); 8.15 (d, 1H, -C16H-, J = 5.7); 9.04 (s, 1H, -N7H-); 9.31 (s, 1H, -N9H-). HPLC-MS: [M + 1]+ found 282.25; calculated value is 282.12.

Compounds from the **II**-series were synthesized according to refs. [9,10].

O-i-Propyl-*N*-(2-hydroxyethylamino)carbamate (**II***a*) was synthesized according to known procedures [35].

Briefly, for the oxamate derivatives, a solution of 1 eq. of *O*-alkyl-*N*-(2-hydroxyethyl)oxamate in dry toluene (15 mL per 2 g of substance) was placed in a round bottom flask equipped with a calcium chloride tube and magnetic stirrer. Then, the solution of 1 eq. 4-methyl phenyl isocyanate in dry toluene (30 mL per 1.5 g of isocyanate) and 2–3 drops of triethylamine were added. The reaction mixture was stirred at room temperature for 15 min, wherein precipitation was formed. The resulting precipitate was filtered off. The product was purified by recrystallization from isopropanol.

*O*-Propyl-*N*-[2-(4-methylphenylaminocarbonyloxy)ethyl]oxamate (**II***b*). 35% yield, mp = 123–125 °C. ^1^H NMR (DMSO-*d_6_*, δ, ppm, J, Hz): 0.92 (t, 3 H, CH3, J3 = 8.0); 1.63 (sextet, 2 H, CH2CH3, J3 = 8); 2.28 (s, 3 H, CH3CHarom); 3.85–4.13 (m, 6 H, CH2O, CH2NH, CH2OCO); 7.01 (d, 2 H, m-CHarom, J3 = 8.0); 7. 14 (d, 2 H, o-CHarom, J3 = 8.0); 9.08 (bs, 1 H, NHCOO); 9.47 (bs, 1 H, NHCOO). ^13^C NMR (DMSO-*d_6_*, δ, ppm): 10.60 (CH3CH2); 20.78 (CH3CH2); 21.73 (CH3Carom); 62.36 (CH2NH); 67.85 (CH2OCONH); 68.36 (CH2OCO); 118.66 (CH3Carom); 131.69 (m-CHarom); 136.85 (o-CHarom); 137.64 (ipso-Carom); 153.74 (CONH); 157.63 (OCNH); 161.03 (OCO). HPLC-MS: [M + 1]+ found 309.30.; calculated value is 309.14.

*O*-Isobutyl-*N*-[2-(4-methylphenylaminocarbonyloxy)ethyl]oxamate (**II***d*). 55% yield, mp = 205–210 °C. ^1^H NMR (DMSO-*d_6_*, δ, ppm, J, Hz): 0.91 (d, 6 H, CH3, J3 = 6.2); 1.96 (septet, 1 H, CHCH3, J3 = 6.4); 2.24 (s, 3 H, CH3CHarom); 3.45 (bs, 2 H, CH2O); 3.99 (d, 2 H, CHCH2O, J3 = 6.2); 4.18 (t, 2 H, CH2NH, J3 = 5.0); 7.08 (d, 2 H, m-CHarom, J3 = 7.7); 7. 33 (d, 2 H, o-CHarom, J3 = 7.5); 9.04 (bs, 1 H, NHCOO); 9.57 (bs, 1 H, NHCOO). ^13^C NMR (DMSO-*d_6_*, δ, ppm): 19.18 (CH3); 20.78 (CH3Carom); 27.58 (CHCH3); 54.67 (CHCH2O); 62.37 (CH2NH); 72.06 (CH2O); 118.69 (CH3Carom); 129.63 (m-CHarom); 131.00 (o-CHarom); 136.94 (ipso-Carom); 153.84 (CONH); 157.69 (OCNH); 161.04 (OCO). HPLC-MS: [M + 1]+ found 323.10; calculated value is 323.15.

The synthesis of the compound **II***c* is described in the literature [9,10].

Compounds from the **III**-series were synthesized according to ref. [19].

Briefly, for the aryl carbamates (**III***a*, *b*, *c*, *d*) synthesis, 1 eq. of 2-hydroxyethyl derivative in a small volume of dry acetonitrile (20 mL per 1 g of substance) was placed in a round bottom flask equipped with a calcium chloride tube and magnetic stirrer. Then, a solution with 1 eq. of the relevant phenyl isocyanate in dry acetonitrile (30 mL per 1 g of substance) and 2–3 drops of triethylamine were added. The reaction mixture was stirred at room temperature for 24 h. The solution was evaporated to dryness, and the residue was recrystallized from methanol and from isopropanol. The precipitate was filtered off and washed with a small amount of cold isopropanol.

2-(2-oxoimidazolidin-1-yl) ethyl-N-phenyl carbamate (**III***a*). 15% yield, mp = 115–117 °C. ^1^H NMR (DMSO-*d_6_*, δ, ppm, J, Hz): 3.17–3.26 (m, 2H, -N-CH2-CH2-cycle); 3.29–3.34 (m, 2H, -N-CH2-CH2-cycle); 3.38–3.45 (m, 2H, -N-CH2-CH2-O-); 4.13–4.18 (m, 2H, -N-CH2-CH2-O-); 6.21 (s, 1H, -NH-C(O)-N-); 6.97 (tt, 1H, CHaryl, *J* = 7.4, 0.9); 7.18–7.32 (m, 2H, CHaryl); 7.39–7.47 (m, 2H, CHaryl); 9.50 (s, 1H, -NH-C(O)-O-). HPLC-MS: [M + 1]+ 250.24; calculated value is 250.27.

2-(2-oxoimidazolidin-1-yl)ethyl-*N*-(3-chlorophenyl) carbamate (**III***b*). 35% yield, mp = 112–114 °C. ^1^H NMR (DMSO-*d_6_*, δ, ppm, J, Hz): 3.19–3.22 (m, 2H, -N-CH2-CH2-cycle); 3.28–3.33 (m, 2H, -N-CH2-CH2- cycle); 3.37–3.43 (m, 2H, -N-CH2-CH2-O-); 4.16 (t, 2H, -N-CH2-CH2-O-, J = 5.1); 6.36 (s, 1H, -NH-C(O)-N-); 7.02 (m, 1H, CHaryl); 7.28 (m, 1H, CHaryl); 7.36 (m, 1H, CHaryl); 7.59 (m, 1H, CHaryl); 9.89 (s, 1H, -NH-C(O)-O-). HPLC-MS: [M + 1]+ 284.20; calculated value 284.07

For the compounds **III***c* and **III***d* see ref. [19].

Briefly, for the aryl ureas (**III***e*, *f*, *g*, *h*) synthesis 1 eq. of amine in dry toluene (50 mL per 4 g of substance) was placed in a three-necked flask with a thermometer, a dropping funnel, and a magnetic stirrer. The mixture was cooled in an ice bath to a temperature no higher than 5 °C. Then, a solution with 1 eq. of the relevant phenyl isocyanate in dry toluene (50 mL per 3.5–4 g of substance) was added dropwise with stirring, keeping a temperature no higher than 5 °C. The reaction mixture was stirred at room temperature for 24 h. The precipitate was filtered off and recrystallized from acetone.

*N*-[2-(2-oxoimidazolidin-1-yl)ethyl]-*N’*-(3-chlorophenyl) urea (**III***f*). 16% yield, mp = 137–138 °C. ^1^H NMR (DMSO-*d_6_*, δ, ppm, J, Hz): 3.12 (t, 2H, -N-CH2-CH2-cycle, J = 5.8); 3.17–3.25 (m, 2H, -N-CH2-CH2-NH); 3.33–3.39 (m, 2H, -N-CH2-CH2-NH-); 6.134 (t, 1H, -NH-C(O)-NH-); 6.17 (s, 1H, -NH-C(O)-N-); 6.87–6.92 (m, 1H, CHaryl); 7.13–7.24 (m, 2H); 7.63 (t, 1H, CHaryl, J = 1.8); 8.66 (s, 1H, -NH-C(O)-NH-). HPLC-MS: [M + 1]+ found 283.29; calculated value is 283.09.

The compounds **III***e*, *g*, *h* are described in the literature [19].

### 4.4. Cell Culture

All cell lines were maintained in a CO_2_ incubator at 37 °C, 95% humidity and 5% CO_2_. The composition of the culture medium for the cells was as follows: MDA-MB-231 (ATCC HTB-26), Bj-5ta (ATCC CRL-4001), and A-375 (ATCC CRL-1619): DMEM, 4 mM L-Gln, 10% fetal bovine serum (FBS), U-87 MG (ATCC HTB-14): MEM, 2 mM L-Gln, 1% non-essential amino acids, 1 mM pyruvate, and 10% FBS; SH-SY5Y (ATCC CRL-2266): 1:1 MEM: F12, 10% FBS, 2 mM L-Gln, 0.5 mM sodium pyruvate, 0.5% non-essential amino acids. The cells were routinely checked for mycoplasma contamination using RT-PCR. All cell media contained 100 U/mL penicillin, 100 µg/mL streptomycin, and 2.5 µg/mL amphotericin B. The cells were passaged using Trypsin-EDTA solution (PanEco, Moscow, Russia), the continuous passaging time did not exceed 40 passages.

Mycoplasma contamination was controlled using the Mycoplasma Detection Kit (Jena Bioscience, Jena, Germany).

### 4.5. Oxidative Stress Induction

For cell viability experiments, the cells were seeded at a density of 30,000 per well of a 96-well plate in 100 µL of the test medium (culture medium with 50 mM HEPES, pH 7.4, and without serum and pyruvate) and incubated for 12 h. After that, a substance solution with or without the toxic agent in 100 µL fresh test medium was added to the medium present in the wells and incubated for 24 h, after which cell viability was measured using the MTT assay. Cytotoxicity was induced by either 100 µM of H_2_O_2_ or 700 µM of CoCl_2_ (from the freshly prepared stock in EtOH).

### 4.6. Cytotoxicity and Proliferation Stimulation

For analysis of cell death induction and ROS generation, the cells were plated in 96-well plates at a density of 1.5 × 10^4^ cells for the cytotoxicity assay and 8000 for the proliferation study per well and grown overnight. The dilutions of the test compounds prepared in DMSO and dissolved in the culture medium (without serum starvation) were added to the cells in triplicate for each concentration (100 µL of the fresh medium with the substance to 100 µL of the old medium in the well) and incubated for 18 h in the case of cytotoxicity and 72 h in the case of the proliferation stimulation. The incubation time was chosen based on the most pronounced differences between the compounds tested. The final DMSO concentration was 0.5%. Negative control cells (100% viability) were treated with 0.5% DMSO. Positive control cells (100% cell death) were treated with 3.6 μL of 50% Triton X-100 in ethanol per 200 μL of the cell culture medium. Separate controls were without DMSO (no difference with the control 0.5% DMSO was found). Depending on the experiment series, the effects of the test substances on the cell viability and ROS production were evaluated using the MTT assay and DCFH-DA, accordingly.

### 4.7. Cell Viability Assay

Cell viability was analyzed using the MTT test [36]. In short, the culture medium was removed from the wells and 75 µL of the 0.5 mg/mL solution of MTT with 1 g/L D-glucose in Earle’s salts was added to each well and incubated for 90 min in the CO_2_ incubator at 37 °C. After that, 75 µL of 0.04 M HCl in isopropanol was added to the MTT solution in each well and incubated on a plate shaker at 37 °C for 30 min. The optical density of the solution was determined using a Hidex Sense Beta Plus microplate reader (Hidex, Turku, Finland) at the wavelength of 570 nm with a reference wavelength of 620 nm.

### 4.8. Apoptosis Assay

Apoptosis level was analyzed using an Apoptosis/Necrosis detection kit (ab176749, Abcam, Cambridge, UK). The cells were seeded at a density of 15,000 per well of a 96-well plate and grown for 12 h. After that, 475 µM of H_2_O_2_ alone or with the peptide was added in 100 µL of the fresh medium to 100 µL of the old medium in the wells and incubated for 1 h at 37 °C in a CO_2_ incubator. After that, the medium was removed, and the cells were stained according to the manufacturer’s instructions using the phosphatidylserine sensor (apoptotic cells, green fluorescence) and membrane-impermeable dye 7-AAD (necrotic cells, red fluorescence). The stained cells were photographed using an inverted fluorescent microscope Nikon Ti-S using a Semrock GFP-3035D filter cube with magnification 100×. For each well, five non-intersecting view fields were captured, and apoptotic cells were counted.

### 4.9. Caspase Activity Assay

The determination of caspase activity was performed using the specific substrates with a fluorescent 7-amido-4-trifluoromethylcoumarin (AFC) label. Cells were seeded into a 96-well plate (7 × 10^4^ cells/well) and incubated overnight. Test compound solutions in the full culture medium were added to the cells without medium change and incubated in a CO_2_ incubator for 4 h at 37 °C. A pan-caspase inhibitor Z-VAD-FMK (80 μM) was used as a negative control. Then, the medium was discarded and 120 µL of the caspase assay buffer (20 mM HEPES, 2 mM EDTA, 0.1% CHAPS, 5 mM dithiothreitol, protease inhibitor cocktail, pH 7.4) was added to the cells. Then, the cells were frozen at −50 °C. After thawing, 120 µL of the caspase substrates Ac-DEVD-AFC (32 μM), Ac-LEHD-AFC (32 μM), and SCP0139 (32 μM) were added to the cell lysates and incubated for 90 min at 37 °C. The released AFC determination was performed using the Hidex Sense Beta Plus microplate reader (Hidex, Turku, Finland) at λ_ex_ = 505 nm, λ_em_ = 400 nm.

### 4.10. ROS Assay

ROS generation was measured using the DCFH-DA dye. The cells were seeded at a density of 60,000 per well of a 96-well plate and grown for 12 h. After that, the cells were treated with the substances in the culture medium for 24. Cells treated with a medium without H_2_O_2_ and substances were used as a control. After that, the medium was replaced with a fresh one with 25 μM of the dye, and the cells were incubated in the CO_2_ incubator at 37 °C for 1 h. After the incubation, the cells were washed twice with Earle’s balanced salt solution, and the fluorescence was measured using the plate reader Hidex Sense Beta Plus (Hidex, Turku, Finland), λ_ex_ = 490 nm, λ_em_ = 535 nm.

### 4.11. Molecular Docking

Ligand structures were obtained from the PubChem database (https://pubchem.ncbi.nlm.nih.gov/, access date 1 May 2022) or prepared manually using Avogadro 1.93.0 software and optimized using the OpenBabel 3.0.0 software (http://openbabel.org/, access date 1 May 2022) [37] using the FFE force field with Fastest descent and dE ≤ 5 × 10^−6^ threshold. Protein structures were obtained from the PDB database (https://www.rcsb.org/, access date 1 May 2022) and optimized using the Chiron service (https://dokhlab.med.psu.edu/chiron/processManager.php, access date 01.05.2022) [38]. Molecular docking was performed using the AutoDock Vina 1.1.2 (http://vina.scripps.edu/, access date 1 May 2022) [39]. To detect possible alternative binding sites and compare the affinities of the ligands for them, the procedure described in the literature [40] was used. As such, molecular docking was performed in two steps: first, we docked each molecule to the whole receptor as one large binding area to locate potential alternative binding sites, then the coordinates of the docking results were clustered and averaged to give the centers of the binding sites. The grid center coordinates are represented in the Table 6. For large proteins, several grid centers were used to cover the whole protein. In all cases, the grid size was 126 × 126 × 126 Å, chosen to cover the whole protein, and exhaustiveness was set to 16. For each ligand, the docking was performed 10 times with different random seeds generating 10 conformations each time. The resulting coordinates were clustered using the OPTICS algorithm [41] from the package scikit-learn [42].

### 4.12. Statistics

All experiments were performed at least in triplicate. Statistical analysis was performed with the GraphPad Prism 9.0 software using ANOVA with the Holm-Sidak or Tukey post-tests; *p* ≤ 0.05 was considered a statistically significant difference.

## 5. Conclusions

In this paper we report the synthesis of some aryl carbamate, pyridyl urea, and aryl urea derivatives with alkyl and chlorine substitutions and tests of their cytotoxic and cytoprotective activity. Aryl carbamates with an oxamate moiety were anti-proliferative for the cancer cell lines tested, while the aryl ureas were inactive. In the cytoprotection studies, aryl carbamates were able to counteract the CoCl_2_ cytotoxicity by 3–8%. The possible molecular targets of the aryl carbamates with oxamate moiety during the anti-proliferative action were the adenosine A2 receptor and CDK2.

The novelty of the research was the screening of the chemically synthesized cytokinin analogs, which have never been characterized for such biological activity before. Although most of the compounds displayed little activity in the most tests, compounds of the series **II** displayed an interesting highly selective antiproliferative capacity. This activity was observed, among others, for the glioblastoma cell line. Given the lack of efficient treatments for this cancer type, such activity could be used in the combined or supporting therapy after additional research.

## Figures and Tables

**Figure 1 molecules-27-03616-f001:**
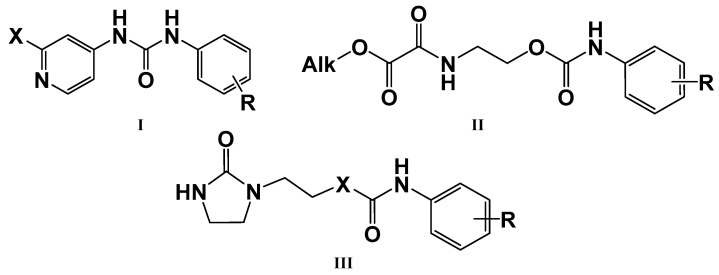
Cytokinin-like compounds.

**Figure 2 molecules-27-03616-f002:**
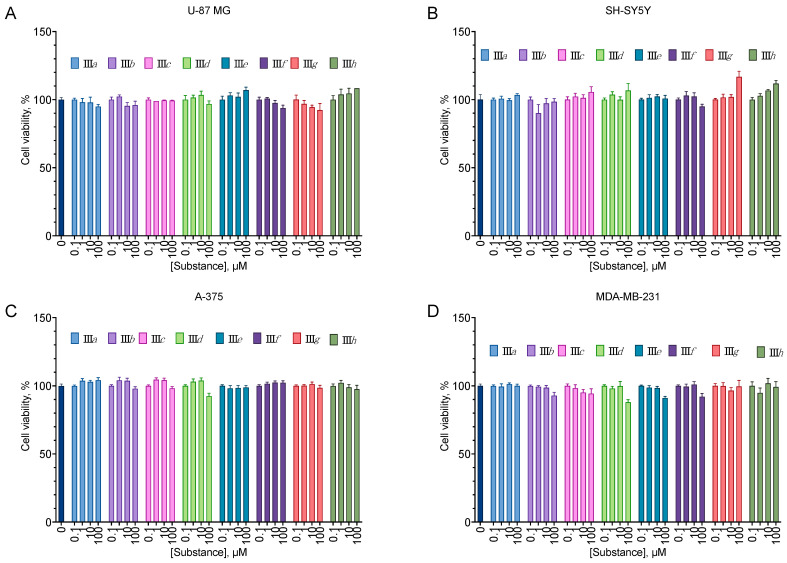
Anti-proliferative activity of the compounds **III***a*–*h* on the U-87 MG glioblastoma (**A**), SH-SY5Y neuroblastoma (**B**), A-375 melanoma (**C**), and MDA-MB-231 carcinoma (**D**) cell lines. Negative control cells (100% viability) were treated with 0.5% DMSO. Positive control cells (100% cell death) were treated with 3.6 μL of 50% Triton X-100 in ethanol per 200 μL of the cell culture medium. 24 h incubation. MTT test data. Mean ± standard error (*n* = 5 experiments).

**Figure 3 molecules-27-03616-f003:**
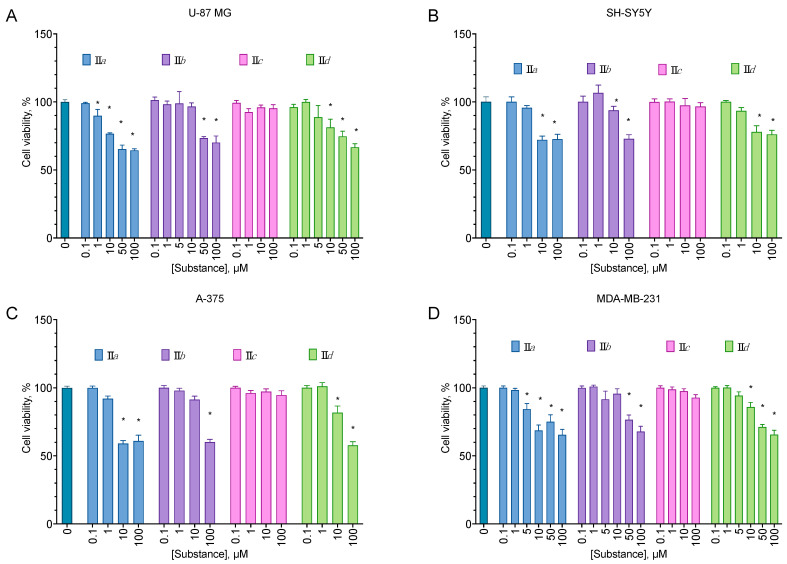
Anti-proliferative activity of the aryl carbamates for the U-87 MG glioblastoma (**A**), SH-SY5Y neuroblastoma (**B**), A-375 melanoma (**C**), and MDA-MB-231 carcinoma (**D**) cell lines. Negative control cells (100% viability) were treated with 0.5% DMSO. Positive control cells (100% cell death) were treated with 3.6 μL of 50% Triton X-100 in ethanol per 200 μL of the cell culture medium. 24 h incubation. MTT test data. Mean ± standard error (*n* = 5 experiments). * Statistically significant difference from the control, ANOVA with the Dunnett post-test, *p* ≤ 0.05.

**Figure 4 molecules-27-03616-f004:**
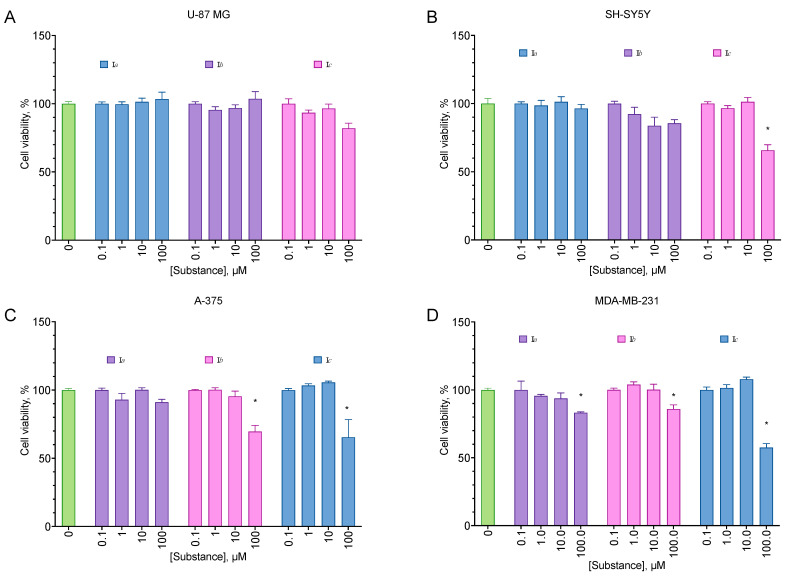
Anti-proliferative activity of the pyridyl urea derivatives for the U-87 MG glioblastoma (**A**), SH-SY5Y neuroblastoma (**B**), A-375 melanoma (**C**), and MDA-MB-231 carcinoma (**D**) cell lines. Negative control cells (100% viability) were treated with 0.5% DMSO. Positive control cells (100% cell death) were treated with 3.6 μL of 50% Triton X-100 in ethanol per 200 μL of the cell culture medium. 24 h incubation. MTT test data. Mean ± standard error (*n* = 5 experiments). * Statistically significant difference from the control, ANOVA with the Dunnett post-test, *p* ≤ 0.05.

**Figure 5 molecules-27-03616-f005:**
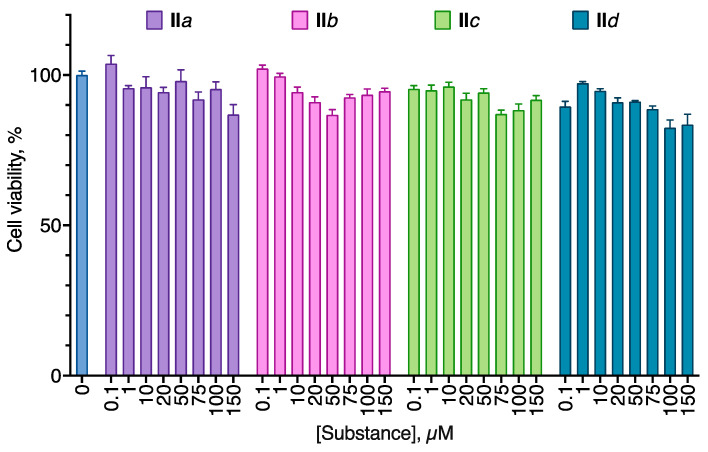
Anti-proliferative activity of the aryl carbamates **II** for the human immortalized fibroblast Bj-5ta cell line. Negative control cells (100% viability) were treated with 0.5% DMSO. Positive control cells (100% cell death) were treated with 3.6 μL of 50% Triton X-100 in ethanol per 200 μL of the cell culture medium. 24 h incubation. MTT test data. Mean ± standard error (*n* = 3 experiments).

**Figure 6 molecules-27-03616-f006:**
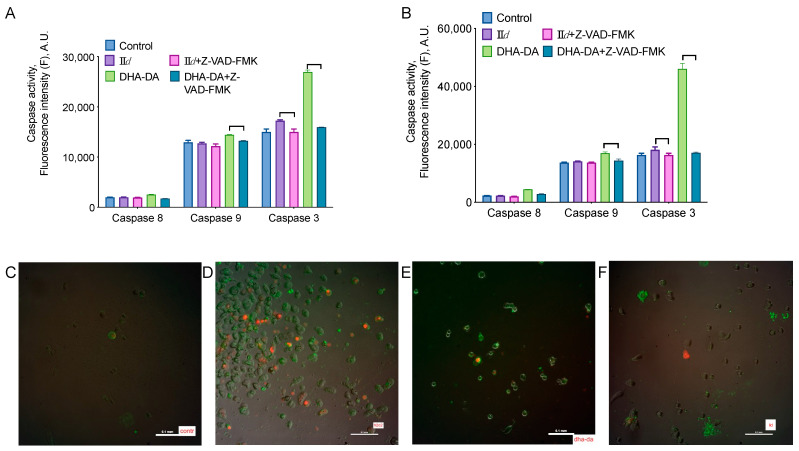
Apoptosis induction by the **II***d* compound for the MDA-MB-231 cell line. Caspases activity after 5 (**A**) and 2 (**B**) h of incubation with 100 µM of GT-04 or 90 µM of apoptosis inductor (N-docosahexaenoyl dopamine, DHA-DA) with or without 80 µM of Z-VAD-FMK. amalgamated data of *n* = 3 experiments. Membrane integrity loss (DNA-binding dye propidium iodide, red) and phosphatidylserine externalization (annexin-FITC dye, green) in the control cells (**C**) and after the treatment with 5 mM of H_2_O_2_ (**D**), 80 µM of DHA-DA (**E**) or 100 µM of **II***d* (**F**).

**Figure 7 molecules-27-03616-f007:**
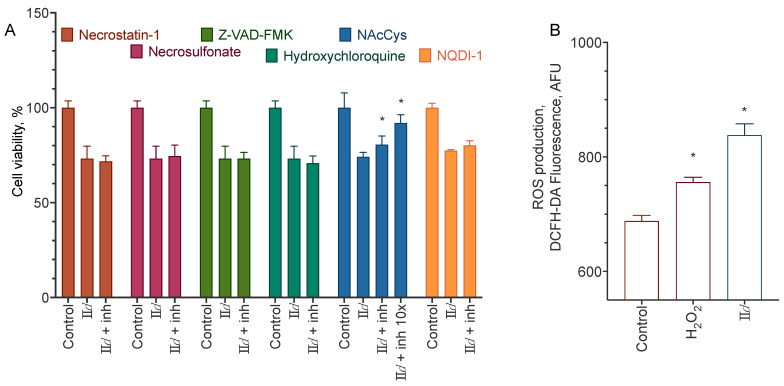
Cell death mechanism of the **II***d* compound for the MDA-MB-231 cell line. (**A**) Effect of necroptosis (necrostatin-1, 100 µM, and necrosulfonate, 1 µM), apoptosis (Z-VAD-FMK, 10 µM), autophagy (hydroxychloroquine sulfate, 1 µM), and ASK1 (NQDI-1, 10 µM) blockers and ROS scavenger (N-Ac-Cys, 0.05 or 0.5 mM + 50 mM HEPES for pH stabilization) on the cytotoxicity of 100 µM of **II***d*. 24 h incubation time, MTT assay data, mean ± standard error, *n* = 3 amalgamated experiments. (**B**), ROS accumulation after the treatment of the cells with 5 mM of H_2_O_2_ or 100 µM of **II***d*. 24 h incubation time, DCFH-DA fluorescence data, *n* = 3 amalgamated experiments. * Statistically significant difference from the control without blocker (**A**) or substance (**B**), ANOVA with the Tukey post-test, *p* ≤ 0.05.

**Figure 8 molecules-27-03616-f008:**
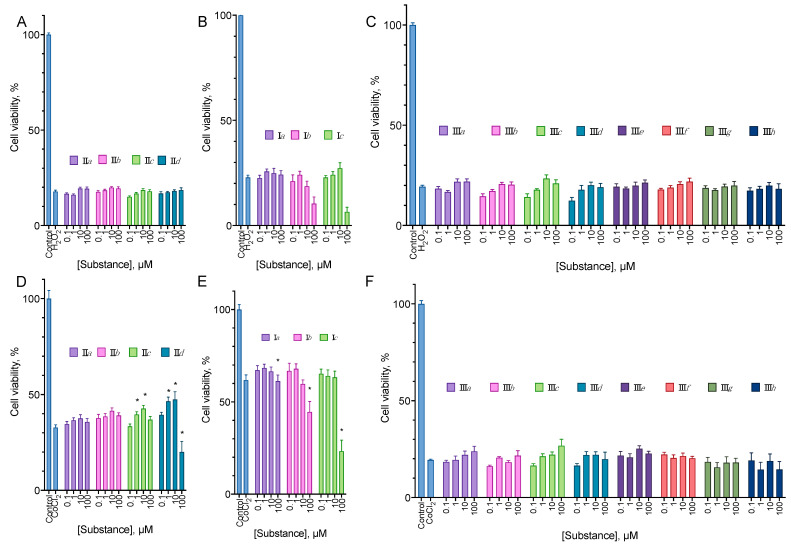
The effect of aryl carbamates **II** (**A**,**D**) pyridyl ureas **I** (**B**,**E**), and EDU analogs **III** (**C**,**F**) on the cytotoxicity of H_2_O_2_ (**A**–**C**) and CoCl_2_ (**D**–**F**) for the SH-SY5Y cell line. 24 h incubation time. MTT assay data, mean ± standard error, *n* = 3 amalgamated experiments. * Statistically significant difference from the control without substance. ANOVA with the Holm-Sidak post-test, *p* ≤ 0.05.

**Figure 9 molecules-27-03616-f009:**
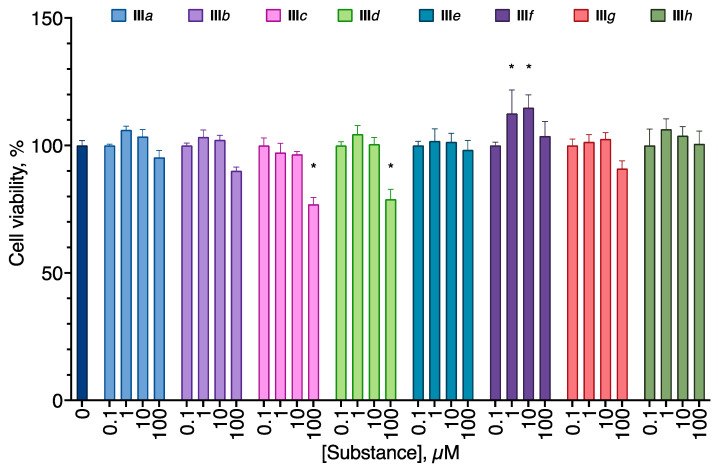
EDU analogs **III** effect on the proliferation of SH-SY5Y cell line in a prolonged incubation. Negative control cells (100% viability) were treated with 0.5% DMSO. Positive control cells (100% cell death) were treated with 3.6 μL of 50% Triton X-100 in ethanol per 200 μL of the cell culture medium. 72 h incubation time. MTT assay data, mean ± standard error, *n* = 3 amalgamated experiments. * Statistically significant difference from the control without substance. ANOVA with the Dunnett post-test, *p* ≤ 0.05.

**Figure 10 molecules-27-03616-f010:**
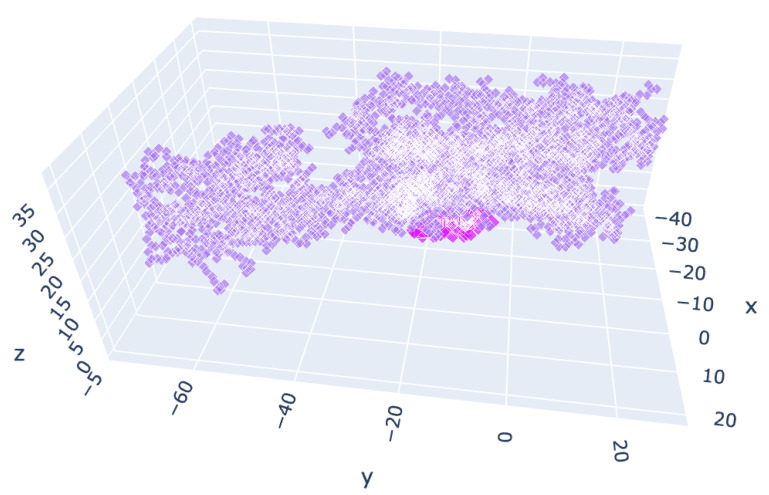
Chosen cluster location for the A2AR variant 5mzj. Violet: centroids of the receptor residues. Purple: centroids of the docked molecules.

**Figure 11 molecules-27-03616-f011:**
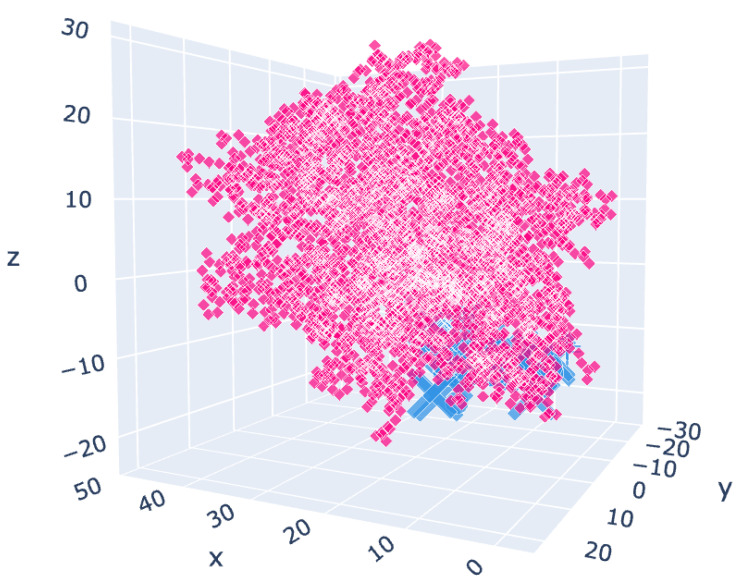
Chosen cluster location for the APRT variant 6hgs. Rose: centroids of the receptor residues. Blue: centroids of the docked molecules.

**Figure 12 molecules-27-03616-f012:**
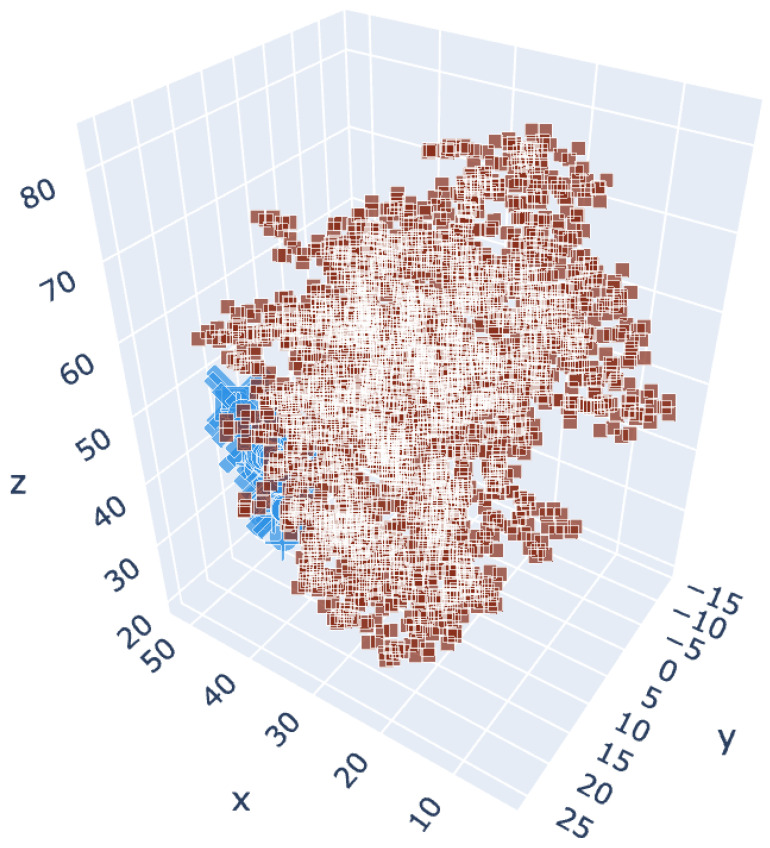
Chosen cluster location for the CDK2 variant 5fp5. Brown: centroids of the receptor residues. Blue: centroids of the docked molecules.

**Table 1 molecules-27-03616-t001:** Structural formulas of synthesized aryl carbamates and ureas.

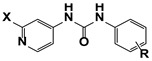			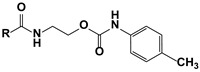		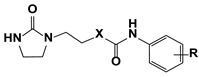			
1-Phenyl-3- (4-Pyridyl) Urea Derivatives			Arylcarbamates		Arylureas			
	R	X		R		X = O, R		X = NH, R
**I** *a*	—	H	**II** *a*	iPrO	**III** *a*	H	**III** *e*	H
**I** *b*	2-CH_3_	Cl	**II** *b*	nPrOC(O)	**III** *b*	3-Cl	**III** *f*	3-Cl
**I** *c*	3-CH_3_	Cl	**II** *c*	iPrOC(O)	**III** *c*	4-Cl	**III** *g*	4-Cl
**I** *d*	3-Cl	Cl	**II** *d*	iBuOC(O)	**III** *d*	3,4-Cl	**III** *h*	3,4-Cl

**Table 2 molecules-27-03616-t002:** Selectivity of the aryl carbamate **II** cytotoxicity for the percent of proliferation decrease at the compound concentration of 100 µM. Incubation time 20 h, MTT assay data, percent of proliferation decrease, mean ± standard error (*n* = 5 experiments). Selectivity was calculated as the anti-proliferative activity ratio for the appropriate cell line to the anti-proliferative activity for the Bj-5ta cell line. ND, not defined.

	A-375		U-87 MG		MDA-MB-231		Bj-5ta
	Proliferation Decrease Mean ± S.E.	Selectivity	Proliferation Decrease Mean ± S.E.	Selectivity	Proliferation Decrease Mean ± S.E.	Selectivity	Proliferation Decrease Mean ± S.E.
**II** *a*	39 ± 4.2	8.4	35.6 ± 1.1	7.7	34.5 ± 3.9	7.4	4.7 ± 2.4
**II** *b*	39.8 ± 1.9	6.1	29.8 ± 4.8	4.5	32.1 ± 3.9	4.9	6.6 ± 1.9
**II** *c*	5.4 ± 3.2	0.5	4.7 ± 2.7	0.4	7.3 ± 2.2	0.6	11.6 ± 2
**II** *d*	42.3 ± 2.6	2.4	33.3 ± 2.6	1.9	34.3 ± 3.2	2.0	17.5 ± 2.6

**Table 3 molecules-27-03616-t003:** Affinity of the synthesized compounds for the adenosine A2 receptor crystal variants. AutoDock Vina data for the most occupied cluster on the protein surface. ND, not present in this cluster. Lower energy means higher affinity.

	A2AR		A2AR + Adenosine		A2AR + Caffeine	
	Energy, Mean ± S.D.	Occurrence Frequency	Energy, Mean ± S.D.	Occurrence Frequency	Energy, Mean ± S.D.	Occurrence Frequency
Adenosine	−5.37 ± 0.42	0.25	−5.05 ± 0.47	0.16	−5.54 ± 0.47	0.1
Caffeine	−4.53 ± 0.09	0.22	ND	ND	−4.35 ± 0.10	0.02
**II** *a*	−4.87 ± 0.38	0.48	−4.60 ± 0.48	0.46	−4.74 ± 0.34	0.32
**II** *b*	−4.80 ± 0.38	0.26	−4.77 ± 0.48	0.31	−4.61 ± 0.41	0.19
**II** *c*	−4.99 ± 0.41	0.3	−4.81 ± 0.45	0.3	−4.71 ± 0.30	0.23
**II** *d*	−4.97 ± 0.51	0.45	−4.79 ± 0.53	0.28	−5.01 ± 0.34	0.2
**III** *a*	−4.26 ± 0.46	0.32	−3.93 ± 0.37	0.33	−4.38 ± 0.53	0.19
**III** *b*	−4.26 ± 0.59	0.32	−4.17 ± 0.51	0.3	−4.41 ± 0.37	0.27
**III** *c*	−4.25 ± 0.50	0.25	−3.91 ± 0.42	0.26	−4.31 ± 0.38	0.13
**III** *d*	−4.28 ± 0.42	0.24	−4.22 ± 0.40	0.44	−4.25 ± 0.48	0.23
**III** *e*	−4.35 ± 0.43	0.23	−4.33 ± 0.43	0.33	−4.38 ± 0.40	0.31
**III** *f*	−4.43 ± 0.48	0.27	−4.29 ± 0.40	0.33	−4.47 ± 0.51	0.18
**III** *g*	−4.58 ± 0.46	0.29	−4.41 ± 0.40	0.43	−4.53 ± 0.37	0.21
**III** *h*	−4.58 ± 0.60	0.26	−4.46 ± 0.48	0.39	−4.36 ± 0.31	0.19

**Table 4 molecules-27-03616-t004:** Affinity of the synthesized compounds for the APRT crystal variants. AutoDock Vina data for the most occupied cluster on the protein surface.

	APRT + GMP		APRT + IMP		APRT + Phosphate	
	Energy, Mean ± S.D.	Occurrence Frequency	Energy, Mean ± S.D.	Occurrence Frequency	Energy, Mean ± S.D.	Occurrence Frequency
GMP	−6.07 ± 0.45	0.18	−6.16 ± 0.23	0.28	−6.62 ± 0.42	0.34
**II** *a*	−4.57 ± 0.70	0.15	−5.14 ± 0.55	0.31	−4.80 ± 0.50	0.14
**II** *b*	−4.91 ± 0.81	0.19	−4.72 ± 0.75	0.19	−4.90 ± 0.1	0.01
**II** *c*	−4.74 ± 0.34	0.12	−5.07 ± 0.60	0.23	−5.19 ± 0.58	0.23
**II** *d*	−4.53 ± 0.68	0.1	−5.41 ± 0.63	0.27	−4.92 ± 0.36	0.17
IMP	−7.43 ± 0.55	0.35	−7.24 ± 0.49	0.28	−7.73 ± 0.72	0.13
**III** *a*	−4.77 ± 0.57	0.23	−4.33 ± 0.40	0.13	−4.64 ± 0.79	0.14
**III** *b*	−4.65 ± 0.55	0.06	−4.40 ± 0.50	0.32	−4.25 ± 0.30	0.14
**III** *c*	−4.60 ± 0.56	0.05	−4.26 ± 0.40	0.19	−4.67 ± 0.50	0.14
**III** *d*	−4.63 ± 0.51	0.07	−4.48 ± 0.60	0.22	−4.64 ± 0.66	0.15
**III** *e*	−4.41 ± 0.36	0.09	−4.44 ± 0.53	0.13	−4.26 ± 0.49	0.07
**III** *f*	−4.53 ± 0.58	0.12	−4.35 ± 0.41	0.23	−4.43 ± 0.32	0.1
**III** *g*	−4.47 ± 0.33	0.14	−4.37 ± 0.23	0.09	−4.39 ± 0.31	0.14
**III** *h*	−4.50 ± 0.17	0.04	−4.58 ± 0.62	0.15	−4.10 ± 0.10	0.05

**Table 5 molecules-27-03616-t005:** Affinity of the synthesized compounds for the CDK2 crystal variants. AutoDock Vina data for the most occupied cluster on the protein surface.

	CDK2		CDK2 + CyclinB	
	Energy, Mean ± S.D.	Occurrence Frequency	Energy, Mean ± S.D.	Occurrence Frequency
**II** *a*	−4.48 ± 0.51	0.18	−4.87 ± 0.32	0.16
**II** *b*	−4.48 ± 0.38	0.14	−5.09 ± 0.31	0.2
**II** *c*	−4.83 ± 0.44	0.19	−5.21 ± 0.45	0.24
**II** *d*	−4.41 ± 0.38	0.12	−5.14 ± 0.47	0.2
SCP2	−4.97 ± 0.44	0.18	−5.68 ± 0.33	0.22
**III** *a*	−4.13 ± 0.38	0.3	−4.89 ± 0.66	0.12
**III** *b*	−3.93 ± 0.35	0.14	−4.51 ± 0.42	0.1
**III** *c*	−3.98 ± 0.43	0.14	−4.27 ± 0.42	0.12
**III** *d*	−4.11 ± 0.41	0.18	−4.59 ± 0.59	0.16
**III** *e*	−4.39 ± 0.49	0.27	−4.88 ± 0.44	0.17
**III** *f*	−4.30 ± 0.61	0.17	−4.86 ± 0.52	0.22
**III** *g*	−4.32 ± 0.46	0.19	−4.68 ± 0.48	0.09
**III** *h*	−4.54 ± 0.31	0.17	−4.91 ± 0.54	0.15

**Table 6 molecules-27-03616-t006:** Grid centers of the docking experiments.

Protein	Configuration Variant	x	y	z
A2AR 5mzj	1	−17.629	−30.760	18.168
	2	−4.629	−50.760	18.168
	3	−17.629	6.760	18.168
A2AR 2ydo	1	−23.602	10.545	−25.256
	2	−23.602	20.545	−25.256
	3	−4.629	−50.760	18.168
A2AR 5mzp	1	−16.417	−40.474	18.316
	2	−16.417	5.474	18.316
	3	−1.417	−50.474	18.316
APRT 6hgs	1	22.572	−3.082	4.313
APRT 6hgr	1	23.667	−7.067	5.057
APRT 6hgp	1	−24.642	0.247	1.919
CDK2 5fp5	1	29.547	4.964	49.678
CDK2 2jgz	1	55.623	20.504	−10.503
	2	38.623	20.504	5.503
	3	38.623	20.504	−30.503

## Data Availability

The data presented in this study are available on request from the corresponding author. The data are not publicly available due to legal issues.

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
