# Peer review of "Anti-Proliferative and Cytoprotective Activity of Aryl Carbamate and Aryl Urea Derivatives with Alkyl Groups and Chlorine as Substituents"

_molecules, 2022, doi:10.3390/molecules27113616_

Round 1

Reviewer 1 Report

Please for the comments see the pdf in attachments

Author Response

Point 1. line 34: N,N’-diphenyl ureas (Figure 1, Structure I) it's incorrect the drawn structure is N-phenyl-N’-(4- pyridinyl)-urea derivatives.

Response 1. The structure was corrected

Point 2. Moreover, pay attention to all the formalisms required for the IUCAC nomenclature, in particular the use of italics when required (for example: N,N’) . The whole paper needs to be revised.

Response 2. The formalisms were corrected throughout the text

Point 3. lines 68, 313, 341, 342, 363 and 377: “described in [7,8]” or other change with described in literature [7,8] or in the same way as described in ref. 7 and 8. Please edit throughout the text.

Response 3. Edited as requested

Point 4. Table 1: it is not easy to interpret. My opinion would be to change the general name of the compounds with the respective codes, for example Ia-d instead of 1-phenyl-3- (4-pyridyl) urea derivatives.

Response 4. The names were changed

Point 5. Lines 80 and 81: the authors reported the dates “All of the compounds from the analogs of EDU series IIIa-h displayed no cytotoxicity for all cell lines tested (Figure 2)”. What is the difference with respect to the data reported in the following paper? Synthesis of New Compounds in the Series of Aryl-Substituted Ureas with Cytotoxic and Antioxidant Activity. Mendeleev Communications 2020, 30, 153–155, doi:10.1016/j.mencom.2020.03.007.

Response 5. Compared to the mentioned paper, in this research new derivatives were obtained, so the structure sets do not overlap. For some of these new compounds, substantial anti-proliferative activity was discovered and additional data on its possible mechanisms were obtained.

Point 6. Anti-proliferative Activity Evaluation. In many reported results it is not indicated whether a negative and/or positive control was used. The authors should report the data in order to verify the activity of the tested compounds.

Response 6. The controls were added to the figures and figure captions

Point 7. 4.2. Synthesized Compounds Characterization. all formalisms are not respected. For example 1H and 13C NMR change with 1H and 13C NMR or DMSO-d6: change with DMSO-d6

Response 7. Corrected

Point 8. Did the authors check the purity of the synthesized compounds? Please clarified.

Response 8. Purity information was added to the text

Point 9. Spectroscopic data. The authors should provide additional spectroscopic data within a “Supplementary materials” document for all new synthesized compounds.

Response 9. The data were added to the Supplementary materials

Reviewer 2 Report

I am sending my review comments to the manuscript Molecules Number- 1714260 entitled: Anti-Proliferative and Cytoprotective Activity of Aryl Carbamate and Aryl Urea Derivatives with Alkyl Groups and Chlorine as Substituents

Comments to the Author

The manuscript prepared by Maxim Oshchepkov and co- workers describe synthetic cytokinine derivatives are a promising group of cytoprotective and anti-tumor agents. In this paper, the authors described synthesis a set of aryl carbamate, pyridyl urea, and aryl urea cytokinine analogs with alkyl and haloid substitutions and tested their antiproliferative activity

1. Section 4 Materials and Methods contains only information about the materials? 2. Section Chemical synthesis describes The preparation of 1-phenyl-3-(4-pyridyl) urea derivatives (I) was carried out according to known methods [16,17][28] 3. Section Description of chemical synthesis - the preparation of 1-phenyl-3- (4-pyridyl) urea (I) derivatives was carried out using known methods [16, 17] [28] and the known compounds were obtained 1a [28], 1b[28], 1c[28], 1d[28] - is the information on the structure new?

4.Conclusion is too short.

In my opinion, conclusion is to short – this should be improuved.

After careful reading of this paper I am of the opinion that the work is suitable to Molecules.

With Regards

Author Response

Point 1. Section 4 Materials and Methods contains only information about the materials? 

Response 1. The data on the materials used for the chemical synthesis were added. Section 4 contains both data on the materials and methods according to the journal rules.

Point 2. Section Chemical synthesis describes The preparation of 1-phenyl-3-(4-pyridyl) urea derivatives (I) was carried out according to known methods [16,17][28] 

Response 2. The description of the synthesis procedure was elaborated

Point 3. Section Description of chemical synthesis - the preparation of 1-phenyl-3- (4-pyridyl) urea (I) derivatives was carried out using known methods [16, 17] [28] and the known compounds were obtained 1a [28], 1b[28], 1c[28], 1d[28] - is the information on the structure new?

Response 3. The structure information was earlier published only in the patent literature and not made available to the public, and thus we decided to provide it

Point 4. Conclusion is too short. In my opinion, conclusion is to short – this should be improuved.

Response 4. The conclusion was extended

Reviewer 3 Report

The manuscript by Oshchepkov and coworkers report the synthesis of several known compounds and a few new derivatives using known synthetic protocols. The authors also are reporting some bioactivity. Specifically, the anti-proliferative and cytoprotective activity of the synthesized compounds. Unfortunately, the compounds are very little or inactive, with some aryl carbamates counteracting the CoCl2 cytotoxicity by only 3-8%. In general, the research design is good but it lask novelty and significance. Also, the abstract, the last paragraph of the introduction, and the conclusion are practically the same. 

Author Response

Point 1. Unfortunately, the compounds are very little or inactive, with some aryl carbamates counteracting the CoCl2 cytotoxicity by only 3-8%. general, the research design is good but it lask novelty and significance.

Response 1. The novelty of the research was the screening of the chemically synthesized cytokinin analogs, which were never characterized for such biological activity before. Although most of the compounds displayed little activity in the most tests, compounds of the series II displayed and interesting highly selective antiproliferative capacity. This activity was observed among others for the glioblastoma cell line. Given the lack of efficient treatments for this cancer type, such activity could be used in the combined or supporting therapy after additional research.

Point 2. Also, the abstract, the last paragraph of the introduction, and the conclusion are practically the same. 

Response 2. The conclusion was extended. The last paragraph of the introduction was shortened.

Round 2

Reviewer 1 Report

please check formalisms in particular in the Supplementary materials

Reviewer 3 Report

The manuscript is much better.